# A New Alloying Concept for Low-Density Steels

**DOI:** 10.3390/ma15072539

**Published:** 2022-03-30

**Authors:** Jiří Hájek, Zbyšek Nový, Ludmila Kučerová, Hana Jirková, Pavel Salvetr, Petr Motyčka, Jan Hajšman, Tereza Bystřická

**Affiliations:** 1COMTES FHT a. s., Prumyslova 995, 334 41 Dobrany, Czech Republic; zbysek.novy@comtesfht.cz (Z.N.); pavel.salvetr@comtesfht.cz (P.S.); petr.motycka@comtesfht.cz (P.M.); teteby@students.zcu.cz (T.B.); 2Regional Technological Institute, University of West Bohemia, Univerzitni 8, 301 00 Plzen, Czech Republic; skal@rti.zcu.cz (L.K.); hstankov@rti.zcu.cz (H.J.); janh@rti.zcu.cz (J.H.)

**Keywords:** low-density steels, kappa phase, dilatometry

## Abstract

This paper introduces a new alloying concept for low-density steels. Based on model calculations, samples—or “heats”—with 0.7 wt% C, 1.45 wt% Si, 2 wt% Cr, 0.5 wt% Ni, and an aluminium content varying from 5 to 7 wt% are prepared. The alloys are designed to obtain steel with reduced density and increased corrosion resistance suitable for products subjected to high dynamic stress during operation. Their density is in the range from 7.2 g cm^−3^ to 6.96 g cm^−3^. Basic thermophysical measurements are carried out on all the heats to determine the critical points of each phase transformation in the solid state, supported by metallographic analysis on SEM and LM or the EDS analysis of each phase. It is observed that even at very high austenitisation temperatures of 1100 °C, it is not possible to change the two-phase structure of ferrite and austenite. A substantial part of the austenite is transformed into martensite during cooling at 50 °C s^−1^. The carbide kappa phase is segregated at lower cooling rates (around 2.5 °C s^−1^).

## 1. Introduction

The alloying concept for low-density steels is simple in principle, but the production itself is usually complicated. The concept of adding aluminium to steel in large quantities has been used for many decades. Initially, efforts were made to replace the chromium and nickel for applications requiring corrosion resistance [1]. Aluminium embedded in the steel lattice expands the lattice parameter and reduces the density [2,3]. The addition of 10 wt% aluminium leads to the formation of ferrite with a cubic surface-centred lattice and a density reduction of 8% [4]. The maximum proportion of aluminium that can be added to steel is 12 wt%. A higher aluminium content in steel leads to the formation of brittle intermetallic phases such as Fe_3_Al and FeAl, which leads to a deterioration in ductility [5]. When considering steels with a higher aluminium content, where phase transformations occur during cooling, it is necessary to compensate for the high aluminium content with austenite stabilising elements such as carbon and manganese to eliminate the ferritic phase [6]. Carbon and manganese allow austenite formation at higher temperatures and allow the properties to be modified by heat treatment. The formation of kappa carbides in these steels instead of cementite is also observed [5,7]. With a suitable choice of morphology, distribution, proportion, and size, they can benefit the mechanical properties [8]. The aluminium content influences their precipitation. Kappa carbides can form during quenching from the dissolution annealing temperature at aluminium contents higher than 10 wt% or during ageing in the temperature range of 550 to 720 °C if the carbon content is less than 1 wt% [9]. A further increase in mechanical properties can be achieved by nano kappa carbides, which, in the case of homogeneous casting, significantly affects mobility and dislocation ordering during deformation [10]. Intergranular carbides have been identified as very unsuitable [11]. A great advantage in designing the chemical composition of the experimental materials is the use of a database allowing equilibrium thermodynamic calculations. Their reliable use is clearly demonstrated in the paper [12]. 

When sufficient manganese is added to ferritic low-density steels, austenitic steels of the Fe-C-Mn-Al type are formed, which use various strengthening mechanisms to improve properties [5]. These include shear band-induced plasticity (SIP) [2], transformation-induced plasticity (TRIP), twin-induced plasticity (TWIP) [13] and microband-induced plasticity (MBIP) [5].

Low density steels are divided into three different categories according to the carbon, aluminium, and manganese content: austenitic steels (0.5 < wt% C < 2, 8 < wt% Al < 12, 15 < wt% Mn < 30), duplex steels (0.1 < wt% C < 0.7, 3 < wt% Al < 10, 5 < wt% Mn < 30), and ferritic steels (wt% C < 0.03, 5 < wt% Al < 8, wt% Mn < 8) [8,14]. An advantage of using low-density steels (LDS) is the possibility of influencing the yield properties by (Fe, Mn)_3_AlC-type precipitates—kappa carbides and B2-type phases rich in Ni and Cu, respectively. Furthermore, MC, M_3_C, M_23_C_6_, and M_7_C_3_ carbides are found in these steels [15]. The calculations show a clear influence of aluminium on the expansion of the ferrite region and the suppression of the austenite region. The austenite region is known to be expanded by C and Mn. Thermodynamic calculations made by [11] show that kappa carbides are formed in Fe-Mn-Al-C systems at temperatures above 600 °C in the case of Al and C contents above 5 wt% and 1 wt%, respectively. M_23_C_6_-type carbides are formed at lower aluminium contents instead of kappa carbides. At 1000 °C, the carbides are stable only if the carbon content is above 1.5 wt% and the Al content above 12 wt% [16,17].

With increasing aluminium content, there is a significant deposition of brittle phases, leading to a marked reduction in toughness. In [11], the effective use of the B2 phase for strengthening by controlling its morphology and dispersion is described. At the same time, the steel was alloyed with 5 wt% of nickel, which is a very effective element for guaranteeing the formation and stability of the so-called B2 phase. However, it should be noted that the cost of nickel is relatively high.

In [18], the concept of LDS with martensitic transformation based on calculations using Thermo-Calc software is presented. The chemical composition of the results from the observed heats contains 8 wt% Al with a carbon compensation of 1.1 wt%. Low-density steels based on Fe-Mn-A-C are suitable materials for car body parts due to their combination of high strength and low density [7,10,14]. Due to higher corrosion resistance, the increase in aluminium content extends their application to structural parts in thermal power plants or the petrochemical industry [10]. A major disadvantage of most of the published papers is that the experimental heats are usually only hundreds of grams or at most a few kilograms. In the case of miniature heats, however, the effect of the relevant segregation during the solidification of the heat is relatively marginal. This fact then significantly changes the material’s behaviour in the case of larger heats. In our experiment, the chemical composition used most closely resembles the alloying concept of Fe-5Al-1C and Fe-6Al-1C. The work focuses on the description of the essential transformation reactions. Virtually all these steels have a microstructure after rolling consisting of ferrite, predominantly lamellar pearlite, and kappa carbides. The authors precisely describe the individual phases or mixtures in [19]. Our work builds on the above-cited works and comes up with an alloying concept that could be successfully applied in production. The most significant use of the proposed steel is for parts operating under high dynamic stress, e.g., connecting rods, pistons, etc. High thermal stability will also be an essential parameter in some similar applications. Our concept assumes the high thermal stability of the structure combined with heat treatment. A benefit should also be higher corrosion resistance, which is inherently typical for aluminium alloy steels. This paper will present the fundamental physical properties and structural characteristics of the Fe-Al-C-based alloying concept of aluminium-alloy steels. These analyses will be the input for the following analysis of the mechanical and technological properties.

## 2. Materials and Methods

### 2.1. Design of the Experiment

JMatPro software v 12.1 (Sente Software Ltd., Guildford, UK) was used in combination with analytical techniques for experimental planning to design a suitable chemical composition for the low-density steels [20]. Variants of the chemical compositions of steels with reduced density were selected using calculation in JMatPro software supplemented by the Design of the Experiment (DOE) method. Initial designs were based on the chemical composition of 54SiCr spring steel. Boron was used to supplement this. Based on these combinations, 180 variations of chemical compositions were proposed. These results were analysed using EDA^®^JM, which enables the evaluation of parameters not calculated in JMatPro software. In further procedures, the influence of nickel and chromium was also included, bringing the number of variants to 1440 by combining chemical compositions. The Ashby coefficient and the cost were used to eliminate chemical composition variants. Because of the weight requirement, the minimum weight for given energy storage (M2) was used to calculate the Ashby coefficient. The coefficient is based on [21]:(1)$$M_{2}=\sigma_{f}^{2}/E\rho$$
where *σ*_f_ is the yield stress of the given material, *E* is Young’s modulus, and *ρ* is the density of the material.

The five best variants were selected according to the Ashby coefficient, density, Young’s modulus, and yield strength. Based on the intersection of these results, 162 variants were selected, varying in carbon, silicon, chromium, nickel, and aluminium content.

The effect of alloying elements (aluminium, carbon, chromium, nickel, silicon, manganese, and boron) on the Ashby coefficient was statistically evaluated. Another selection criterion was the A_3_ temperature. For practical reasons, 1050 °C was chosen as the maximum A_3_ temperature. The price per ton also further eliminated other options. Ten variants were selected which had an Ashby coefficient higher than four. In addition, emphasis was also placed on the M_f_ temperature, which was prescribed higher than 120 °C. Moreover, we compared the yield strength, tensile strength and density of the variants. From 5 to 7 wt% of aluminium was selected.

### 2.2. Casting and Hot Rolling of Experimental Steels

The production of the experimental material was carried out in a vacuum induction furnace model TN—00-361, Prvni Zelezarska Kladno, Czech Republic in COMTES FHT (Figure 1). The experimental heats were prepared by pouring into casts with dimensions of *D* = 110 mm, *L* = 500 mm. The ingots prepared in this way had a mass of 50 kg (Figure 2). There is such a high amount of aluminium in the steel that it reliably acts as a deoxidising agent. Massive de-oxidation with aluminium produced a large amount of slag which has to be collected during the melting process. Furthermore, the process is characterised by the bubbling of argon. The ingots were chip-machined to their final dimensions of *D* = 97 mm, *L* = 457 mm.

The chemical composition of all three cast heats was determined using a Q4 TASMAN optical emission spectrometer, Bruker AXS GmbH, Karlsruhe, Germany (Table 1).

Samples were cut from the ingots for metallographic analysis. The specimens were prepared by conventional mechanical grinding on SiC foils in several steps up to a grit size of 1200. This was followed by polishing on cloths with a diamond suspension in steps of 9, 3, and 1 μm. The structure was etched with 3% Nital. (Lach-Ner, s.r.o., Neratovice, Czech Republic). Microstructures were documented using an Olympus BX 61 light microscope (Olympus, Shinjuku, Tokyo, Japan) and a Crossbeam Auriga electron microscope with an FEG cathode (Zeiss, Oberkochen, Germany).

## 3. Results

### 3.1. Investigations of the Microstructure after Casting and Hot Rolling

As a result of significant segregation during solidification, the structure in the cast state had a characteristic dendritic arrangement. This can be observed in Figure 3, where the bright spots correspond to the ferrite phase. The ferrite forms the major and minor axes of the dendrites in the structure. The brown areas correspond to the eutectoid mixture that fills the interdendritic spaces. The kappa phase is the most strongly etched feature in the image (black particles). Figure 4, Figure 5 and Figure 6 also show the morphology of the eutectoid (P—pearlite) and kappa phase (κ) with an acicular shape inside the ferrite grains was detected. The arrangement and morphology suggest that this phase was formed at the phase interface and grew inwards towards the ferrite grains.

The structures consisted of identical phases or structural components in all castings. Only their amount in the total volume differed, where the amount of ferrite and kappa phase increased with increasing aluminium content. This is in agreement with [4].

In the next step, the machined ingots were hot rolled. The rolling temperature was 1150 °C, and the rolling took place at 900 °C. Reverse rolling produced a strip with a thickness of 14.4 mm and a width of 200 mm (Figure 7). Deformation in each run ranged from 18% to 23%. Cooling was carried out in air. The microstructure was documented after rolling. Figure 8 shows the microstructure of heat B (6 wt% Al).

Figure 9 and Figure 10 show the microstructure in the longitudinal direction. There is an intense texture of the grains in the rolling direction. The ferrite grains are separated by grains of the kappa phase. Furthermore, pearlite with lamellar morphology is present in the structure. Depending on the aluminium content of the material, a higher amount of kappa phase was segregated during cooling. It is particularly evident when comparing the A (5 wt% Al) and C (7 wt% Al) heats, where the increased amount of the kappa phase is already significant. Due to the significant inhomogeneities in the structure after casting, homogenisation annealing was included, reducing the differences in chemical composition within the micro-volume. The annealing temperature was 1200 °C, soaking time 3 h, with cooling in a furnace. By analysing the image, we could calculate the amount of ferrite in the structure. It understandably increases with the amount of aluminium in the structure, as indicated in Table 2.

The volume fraction of retained austenite was determined by X-ray diffraction phase analysis on a BRUKER D8 DISCOVER (Bruker AXS GmbH, Karlsruhe, Germany), Cu radiation source with wavelength Kα1 = 0.15406 nm, diffractogram measured in the range 30–120° 2 Theta, measurement step size 0.025°, time/step 0.5 s, determination of retained austenite in the range 30–105°2 Theta (α(101); (200); (211); (202); γ (111); (200); (202) + (311) and (222)).

Dilatometric measurements were performed using both the classic and quenching dilatometer. Slow heating experiments at 3 °C/min were performed with the DIL L75 PT dilatometer (Linseis, Selb, Germany) on cylinders with a diameter of 5 mm and length of 20 mm under argon protective gas. Cooling experiments were performed with Linseis RITA L78 quenching dilatometer on cylinders with a diameter of 4 mm and a length of 10 mm under inductive heating and helium cooling gas. Both dilatometers are horizontally arranged, equipped with quartz push rods and LVDT (linear variable differential transformer) length-change sensors. The specimens were held with a contact force of 0.3 N. The temperature at the sample surface was measured with a welded K-type thermocouple.

Density at room temperature was measured by immersion and was calculated as:(2)$$\rho =\rho_{w}\frac{m}{\Delta m}$$
where *ρ*_w_ is the density of water, *m* is the weight of the specimen in air, and Δ*m* is the weight change due to buoyancy when the specimen is submerged.

### 3.2. Thermophysical Property Measurement

#### 3.2.1. Determination of Transformation Temperatures

Determining the transformation temperatures by dilatometry was crucial for this research. Heating at a rate of 3 °C/min to a temperature of 1200 °C was chosen as the first dilatometric measurement. It is a standard rate that is suitable for determining critical points.

Figure 11 shows the dilatometric curves for heats A, B, and C. Especially for heat A with 5 wt% of Al, there is a clear Curie point at temperatures between 600 °C and 650 °C. Its distinctness decreases with increasing aluminium content. Further significant changes occur at temperatures above 900 °C. Above this temperature, the austenitisation and decomposition of the kappa phase and any additional carbide phases (if present in the respective heat) gradually occur. Both processes coincide with and, in addition, have opposite effects on the volume change. It is therefore difficult to observe the exact end of the decomposition of the kappa phase or the dissolution of other carbides by observing the dilatometric indications. The decomposition of the kappa phase plays a significant role, especially for heats with 7 wt% Al (heat C) in the structure. 

Comparing the individual heats, it is clear from Figure 11 that the temperatures of the onset of the phase transformations increase slightly with increasing aluminium content. It is true both for the beginning of austenitisation and for the dissolution of the carbide phases in austenite. In the case of steel containing 5 wt% aluminium (heat A), the beginning of this mixed reaction was above 900 °C. Austenitisation, in this case, terminates completely at 1100 °C. As mentioned above, the decomposition of the kappa phase is accompanied by an increased volume on the dilatometric curve. In the case of heat B (6 wt% Al), the reaction starts above 900 °C (about 910 °C); in the case of heat C (7 wt% Al) at about 960 °C. In all three cases, it can be concluded that the reaction that dissolves the carbides is slightly delayed beyond the onset of the pearlite–austenite reaction.

This indication occurs in all the observed heats. In order to clarify the reactions visible on the dilatometric curve, an experiment was performed with heating only to a given reaction temperature and then quenching in water. After reaching temperatures of 900 °C, 925 °C, 950 °C, 1000 °C, and 1025 °C, the dilatometric measurements were interrupted. The sample was cooled rapidly, metallographic sections were made, and the microstructure was documented. This experiment was carried out for heats A and C.

The initial structure of heat A (after rolling and homogenisation annealing) consisted of a eutectoid mixture—predominantly lamellar pearlite—and ferrite, which contains a tiny proportion of carbide phases (Figure 12). At a temperature of 900 °C, the first austenitic phase islands were already evident in the structure. Furthermore, there were carbide grain particles which are segregated within the ferrite grains. At 900 °C, the pearlite and kappa phases are dissolved mainly in the solid solution. However, it should be noted that in the case of the steel with 5 wt% aluminium, the kappa phase content in the initial structure was very low. There is a re-dissolution of the carbide particles within the ferrite grains with increasing heating temperature. The austenite grains also grow. In dilatometric measurements taken at 1025 °C, a structure containing only ferrite and austenite was visible. The carbides were already almost completely dissolved at this temperature. Thus, this technological process can be used to obtain a carbide-free structure consisting of ferrite, martensite, and retained austenite from heat A (5 wt% Al).

Using metallography, it was found that austenitisation shifts to higher heating temperatures in the steel with 7 wt% aluminium (heat C). The individual microstructures of heat C (Figure 13) show that no phase transformations occur when heating to 900 °C; only changes in the morphology of the carbides in the pearlite. A higher volume of the eutectoid phase into austenite is decomposed at 1000 °C. It is also evident from these images that the decomposition of the kappa phase occurred above 950 °C and again more or less simultaneously with the ongoing austenitisation. All the carbide phases were dissolved when heated to 1025 °C. After rapid cooling, the structure again consisted of ferrite, martensite, and retained austenite. The proportion of martensite, in this case, is clearly smaller than in heat A. It is clear that even with high-temperature heating, the structure is not entirely austenitic, which then also determines the proportion of martensite or retained austenite.

Based on this knowledge, suitable quenching temperatures can be determined. Austenitisation temperatures were determined in the range from 1050 °C for heat A (5 wt% Al) to a maximum of 1100 °C for heat C (7 wt% Al).

#### 3.2.2. Dilatometric Measurements during Cooling

A dilatometric analysis (Figure 14) with a cooling rate of 50 °C/s was conducted in the next step. An austenitisation temperature of 1100 °C with a soaking time of 35 min was used. The cooling rate of 50 °C/s corresponds to the cooling rate in medium-speed quenching oil. We assume standard heat treatment for this kind of structural steel, which means quenching—mostly in oil—and tempering. It is clear from the dilatometric records that the 50 °C/s rate can still be considered subcritical. The pearlitic nose is evidenced by both the dilatometer records and the accompanying metallographic images in Figure 15. The proportion of transformed pearlite decreases with increasing aluminium content. Thus, it is clear that with increasing aluminium content, the onset of transformation shifts to lower cooling rates and longer times. The pearlitic transformation in the experimental heats takes place at slightly higher temperatures than is typical for conventional structural steels, at temperatures above 800 °C.

In the case of steel with 5 wt% Al (heat A), the onset of the martensitic transformation is at temperatures between 230 °C and 240 °C. There is a shift towards lower temperatures for heat B with 6 wt% aluminium (approx. 170 °C). The influence of aluminium is therefore very significant here.

In the case of steel with 7 wt% Al (heat C), the martensite start temperature is no longer evident on the dilatometric curve. There are two reasons for this. The first is a higher amount of stable ferrite. The second is the low proportion of martensite formed due to the higher proportion of retained austenite in the structure after fast cooling.

The microstructures of all the heats formed during the cooling of the dilatometric samples at 50 °C s^−1^ are shown in Figure 15. The microstructures after treatment with a cooling rate of 50 °C s^−1^ consisted of untransformed ferrite and massive areas of martensite, with large amounts of retained austenite remaining between the martensite plates. During the cooling of the dilatometric samples, subcritical cooling occurred, and the nose of the pearlitic transformation in the CCT diagram was intersected. The differences between the heats are in the amount of transformed pearlite, where the amount decreases with increasing aluminium content. Another difference is the decreasing proportion of the transformed structure—martensite—with increasing aluminium content, which contributes to the stability of the ferrite.

X-ray diffraction was used (Figure 16) to evaluate the proportion of retained austenite (RA) in the dilatometric samples processed according to the curves in Figure 14. The amounts of RA in the structure for each heat are shown in Table 3.

#### 3.2.3. Dilatometric Measurements during Tempering

The third set of dilatometric measurements was dedicated to reactions during tempering (Figure 17). First, austenitisation was accomplished at 1100 °C with a soaking time of 35 min. This was followed by cooling at a rate of 50 °C/s. The tempering dilatometric curve was measured with a heating rate of 3 °C/min. Carbide precipitation was evident from the measured curves. Dilatometric indications are more visible in the case of the heats with lower aluminium contents. This can be attributed to the amount of martensite formed during quenching, which is highest in the case of heat A (5 wt% Al). The formation of transit carbides starts at temperatures above 100 °C, and more significantly at about 250 °C. Transit carbides have a predominantly plate-like morphology; their growth is linked to characteristic crystallographic directions (Figure 18). The formation of massive carbides takes place between 580 °C and 700 °C. The dilatometric expression of the formation of these massive carbides is very pronounced in all the heats studied, irrespective of the aluminium content.

In order to quickly demonstrate the individual processes during tempering, metallographic sections were taken from samples terminated at 300 °C using SEM (heat A, Figure 18).

The dilatometric curve shows a reaction when tempering to 700 °C (Figure 17). Light microscope images (Figure 19) show carbide precipitation and the original martensitic plates. Their size and orientation are still apparent. It is also possible to identify the precipitation of fine carbides within these martensitic plates.

On the dilatometric curves (5 wt% Al and 6 wt% Al), a slight increase in volume can be observed from about 400 °C. This may be related to the decomposition of retained austenite.

### 3.3. EDS Analysis

EDS spot analysis of each phase was performed for all heats to identify their chemical compositions. A summary of the results is shown in Table 4. All were carried out in the initial state after homogenisation annealing.

The results show that the aluminium content in the ferrite naturally increases with increasing nominal aluminium content in the heat. Therefore, it can also be assumed that the properties of delta ferrite will not significantly differ from one heat to another. The aluminium content in kappa carbides is elevated compared to the solid solution and ranges between 7.5 wt% and about 12 wt%. The aluminium content in the eutectoid mixture was observed to be slightly lower compared to the other phases. This phenomenon is quite interesting. It shows that delta ferrite dissolves more aluminium than austenite during the solidification of the heat.

In contrast, the chromium content of the eutectoid increases only slightly. There is a significant increase in the chromium content in the eutectoid in heat C. It can be concluded that the aluminium content of the individual phases is not dramatically different and more or less follows the increasing aluminium content of the individual heats.

The EDS analysis in Figure 20 shows the increased aluminium content in the kappa phase compared to the matrix. Some other chromium-rich carbide species are present in places in the eutectoid mixture. Their morphology is predominantly plate-like.

### 3.4. Density Measurement

As the primary benefit of these types of steels is their lower specific gravity, their density was measured. Figure 21 shows the results from the most commonly used hydrostatic method. When compared with standard steels, the density decreased by about 8% in the case of the 5 wt% aluminium heat (A), 10% in the case of the 6 wt% aluminium heat (B), and 11% in the case of the 7 wt% aluminium heat (C).

Gravimetric measurements were also taken as a control. Using this method, the densities were 0.02 g cm^−3^ lower than the results from the hydrostatic method in all cases.

## 4. Discussion

One of the main objectives of the steels developed here was to reduce the density of the material for the potential production of lightweight parts. The alloying concept allows a density reduction of up to 11% compared to conventional steel types. The results confirmed the partial conclusions reported in the literature [6,11,18]. 

The experiment has provided new insights into the behaviour of steels with relatively high aluminium contents. The heats with 5 to 7 wt% aluminium are complex, multiphase systems. Depending on the final heating temperature or the rate of cooling from the relevant temperature, the structure contains ferrite, pearlite, kappa carbides with high aluminium content, martensite, and other types of quenching structures with retained austenite. The amount of ferrite in the initial state after casting, rolling, and homogeneous annealing is approximately 30 to 50%, depending on the amount of aluminium. Therefore, it is clear that in this sense, aluminium is a ferrite-forming element.

### 4.1. Dilatometry

Curie point temperatures were identified from the dilatometric curves. This indication is very pronounced. The temperatures vary from 615 °C to 635 °C depending on the aluminium content in the steel. The individual reactions were described, and the austenitisation temperatures were determined based on the dilatometric curve. The range is from 1050 °C for heat A (5 wt% Al) to 1100 °C for heat C (7 wt% Al). It was found that with increasing aluminium content, the critical points, and thus especially the onset of austenitisation, shift to higher temperatures. It is clear from metallographic analyses that austenitisation does not occur in the entire material volume. The ferrite formations remain stable even at high temperatures. At the same time, it is evident from the metallography that the onset of austenite formation occurs first at the expense of the eutectoid mixture, while the dissolution of the kappa carbides is slightly delayed until part of the structure has already been transformed to austenite. It is particularly evident in heat C, where the proportion of kappa carbides is significant.

Dilatometric curves during the “rapid” cooling from 1100 °C show that the cooling rate of 50 °C/s is already subcritical and that fine pearlite forms along the boundaries of the original austenitic grains. The martensite start temperature decreases depending on the aluminium content of the steel. A difference of 1 wt% in the heat will cause the M_s_ temperature to drop by more than 60 °C. Following the decrease in M_s_ temperature, a significant increase in retained austenite content can be observed with increasing aluminium content in the steel. It increases the proportion of ferrite in the structure. Conversely, the balance of austenite formed decreases. The austenite thus formed contains a high amount of carbon, significantly affecting the decreased temperature, M_s_.

Significant dilatometric indications also accompany the decomposition of martensite during tempering. It is particularly evident in the case of heat A (5 wt% Al), which is due to it having the highest martensite content in the structure. These indications are associated with the precipitation of transition carbides during tempering. These carbides are abundant, have a plate morphology, and are also seen to be present in a crystallographically clear direction. The formation of these carbides ends at about 300 °C. At this temperature, it is also possible to observe the presence of retained austenite in some places.

As mentioned above, dilatometric indications of transition carbide formation were not observed for the C (7 wt% Al) heat. Here, it can be assumed that the main reason for this is the low proportion of martensite formed in the structure during the decomposition. It is not possible to find these changes dilatometrically. From the dilatometric curves, it can be concluded that the decomposition of the retained austenite occurs at temperatures above 400 °C.

The behaviour of the retained austenite in the observed melts is interesting. After quenching, X-ray diffraction measurements show that heat C, with the highest aluminium content, contains significantly more retained austenite than heat A and heat B, with lower aluminium contents. A high retained austenite content is detected here even though heat C had the lowest austenite phase content at austenitisation. The retained austenite is probably the most stabilised in this heat, even during the tempering process. This is due to the high carbon content of austenite. While the dilatometric tempering curves for heats A and B show a dissolution reaction of the retained austenite, this reaction is not evident for heat C.

### 4.2. EDS

From the EDS analysis, it is clear that a relatively large amount of aluminium is dissolved in the ferrite. The content here is higher than in the eutectoid formed from austenite. The Al content of the ferrite varies depending on the aluminium content of the heat. The highest amount of Al is contained in the kappa phase. However, it does not exceed 13 wt% on average in any measurement. From the attached maps of the chemical composition, it can be concluded that the eutectoid also contains some other carbides with higher amounts of Cr which are located mostly along the grain boundaries of the eutectoid.

From the individual analyses of the phase composition, critical temperatures, etc., it is clear that for steel alloyed with 7 wt% aluminium, it would be advisable to increase the carbon content of the steel so that high amounts of ferrite can be avoided. Further heat treatment processes will be carried out based on this study and, in addition to material analyses, the mechanical properties of the processed experimental steels will be measured.

## 5. Conclusions

This paper evaluated the conditions at different processing stages on three LDS with aluminium contents of 5 to 7 wt%. These were the conditions after rolling, quenching, and tempering. In addition to metallographic analysis, dilatometric analysis was used to understand the different phase transformations during processing.

The reduction in specific gravity was up to 11% for the heat with 7 wt% Al.

The very strong effect of ferrite-forming elements is evident in the experiments, especially aluminium and chromium. We come to the significant conclusion that even at very high austenitisation temperatures exceeding 1100 °C, it is impossible to leave the two-phase structure of ferrite and austenite. The amount of ferrite is 30 to 50% of the total volume. Therefore, the alloying concept stabilises the high amount of ferrite in the structure.

The carbide kappa phase is also precipitated at lower cooling rates corresponding to cooling in the air after casting or hot rolling (about 2.5 °C s^−1^). It increases with the aluminium content of the steel. A significant conclusion is that the negative kappa phase in the structure can be eliminated with “rapid” cooling from a sufficiently high heating temperature above 1025 °C. A substantial part of the austenite is transformed into martensite during cooling at 50 °C s^−1^, corresponding to oil quenching. Part of the austenite in the structure remains stabilised.

It can be concluded that the increased aluminium content shifts the martensite start temperature to lower temperatures. This shift is based on the fact that, depending on the higher aluminium content in the heat, the amount of ferrite (as mentioned above, aluminium contributes significantly to the stability of ferrite) increases at the expense of the eutectoid phase or subsequent austenite. Logically, the amount of carbon dissolved in the austenite increases as its proportion decreases. In addition, it should be noted that the quenching temperatures of steels with high aluminium content are relatively high, allowing a rather large amount of carbon to dissolve in the austenite.

During tempering, transition carbides of a plate-like character are formed. These carbides form at temperatures up to about 300 °C—subsequently, the retained austenite decomposes above 400 °C.

## Figures and Tables

**Figure 1 materials-15-02539-f001:**
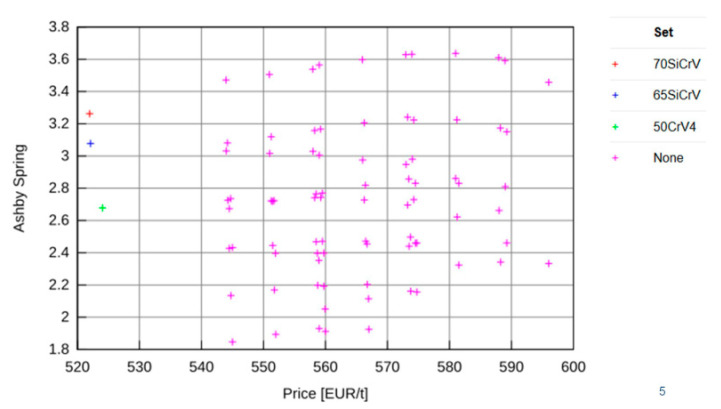
Modification of nickel and chromium-free alloys as a function of the Ashby coefficient to material cost [22].

**Figure 2 materials-15-02539-f002:**
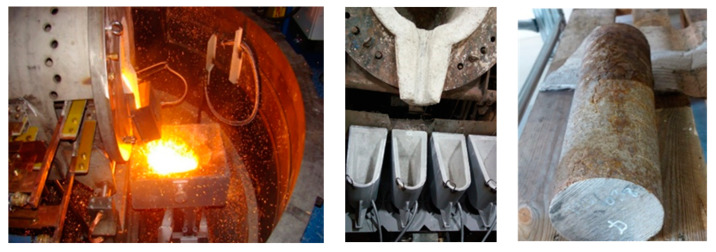
Preparation of experimental material.

**Figure 3 materials-15-02539-f003:**
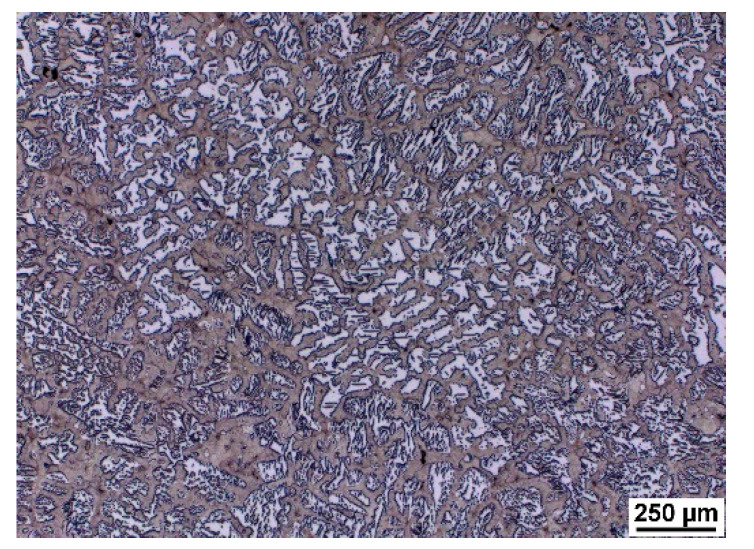
Casting structure of heat C.

**Figure 4 materials-15-02539-f004:**
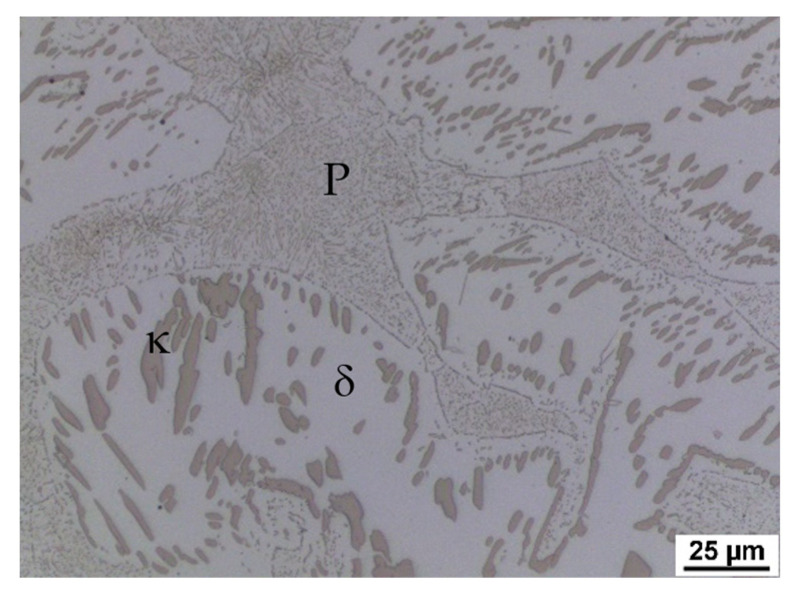
Casting structure of heat C. P—pearlite, κ—kappa phase, δ—ferrite.

**Figure 5 materials-15-02539-f005:**
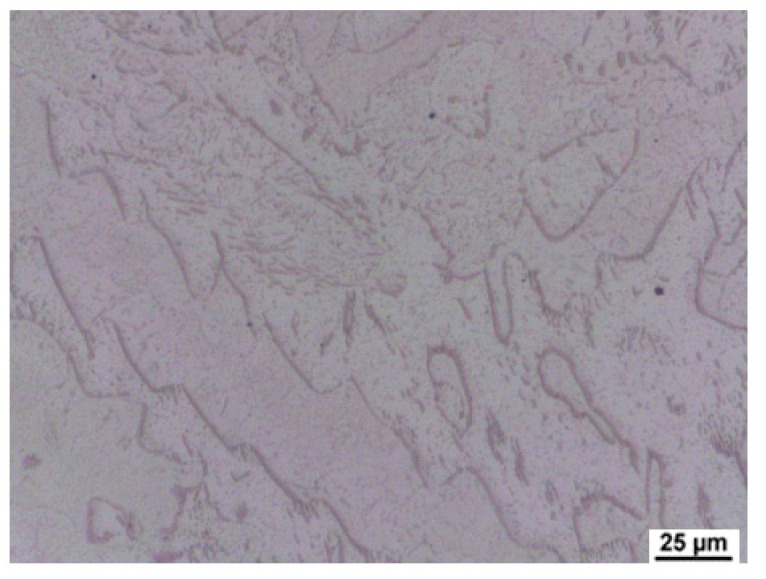
Casting structure of heat A.

**Figure 6 materials-15-02539-f006:**
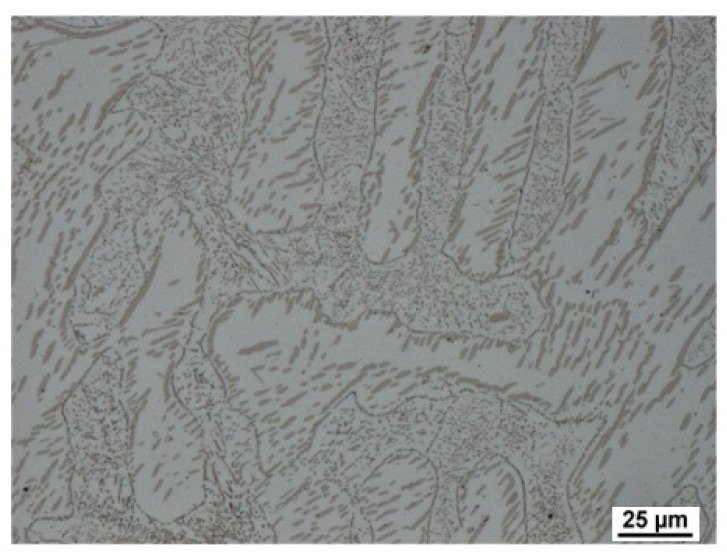
Casting structure of heat B.

**Figure 7 materials-15-02539-f007:**
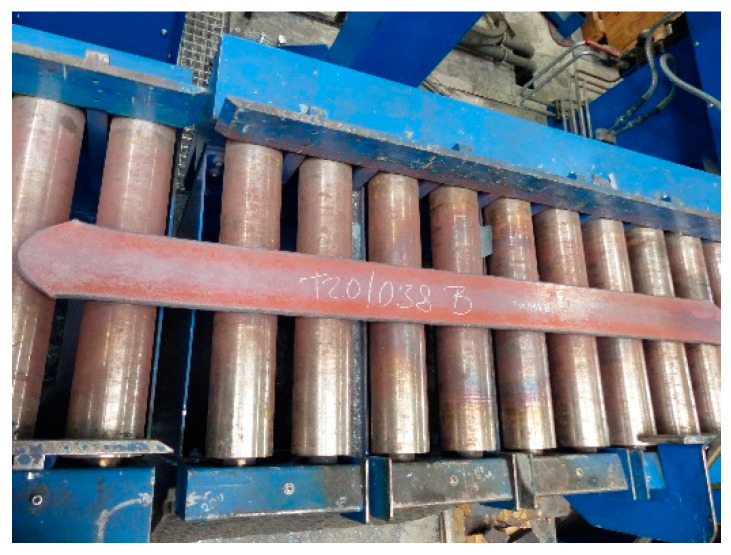
Hot rolled plate. Heat B.

**Figure 8 materials-15-02539-f008:**
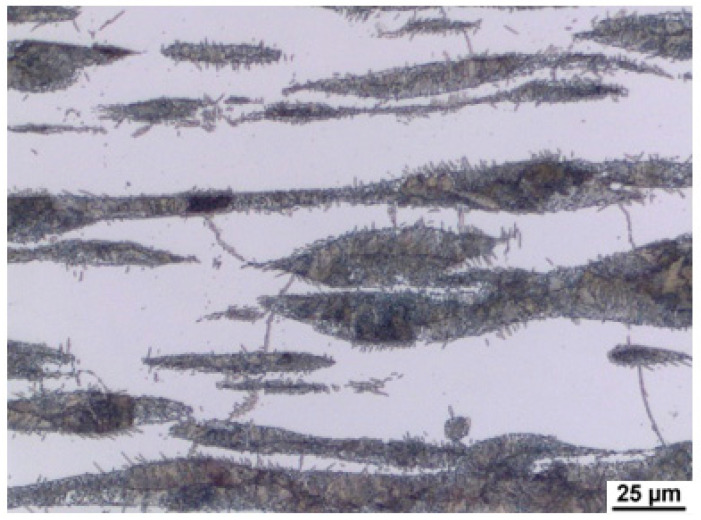
Microstructure of the hot-rolled sheet. Heat B. Longitudinal direction.

**Figure 9 materials-15-02539-f009:**
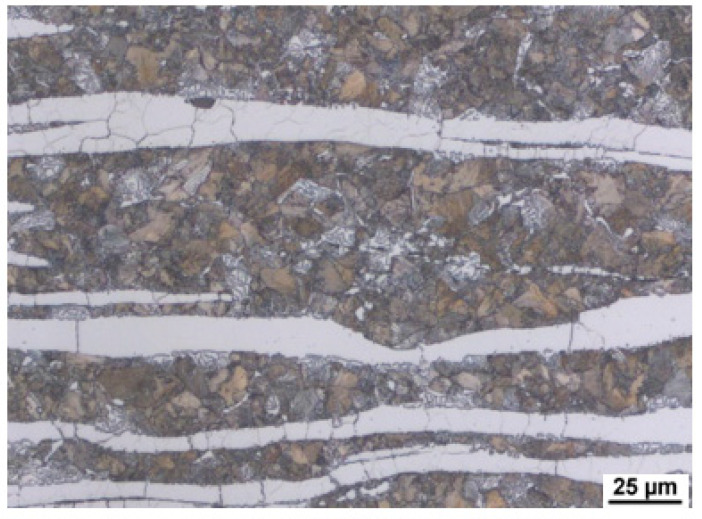
Microstructure of the sheet. Heat A. Longitudinal direction.

**Figure 10 materials-15-02539-f010:**
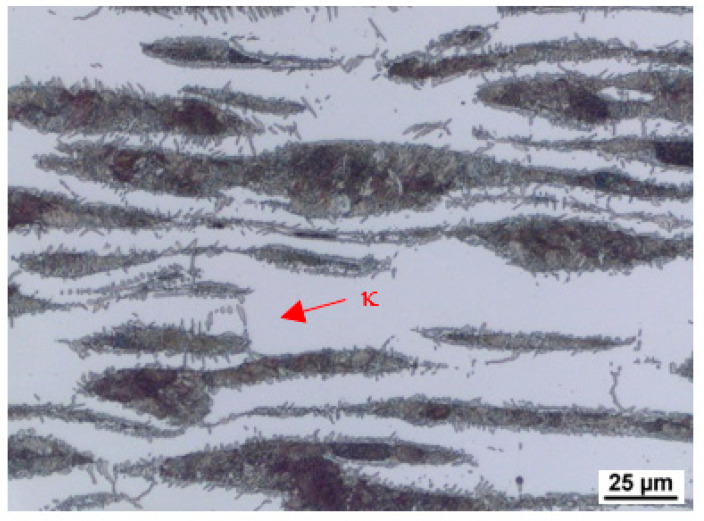
Microstructure of the sheet. Heat C. Longitudinal direction. κ—kappa phase.

**Figure 11 materials-15-02539-f011:**
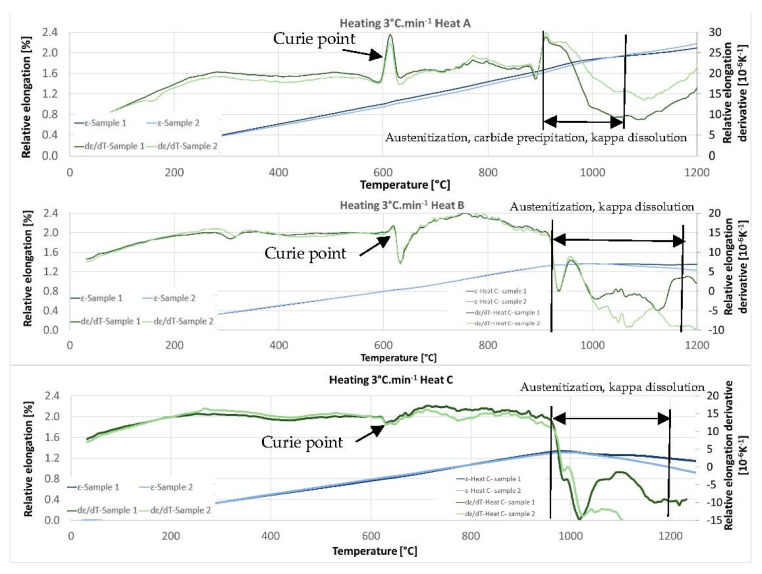
Dilatometric curve—heating 3 °C/min.

**Figure 12 materials-15-02539-f012:**
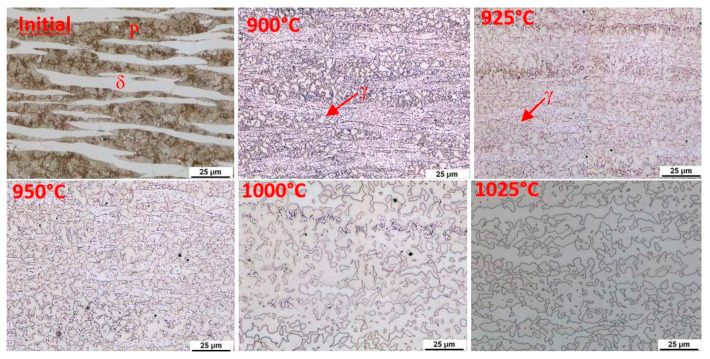
Microstructure of the initial state after different austenitisation temperatures, heat A (5 wt% Al). P—pearlite, δ—ferrite, γ—austenite.

**Figure 13 materials-15-02539-f013:**
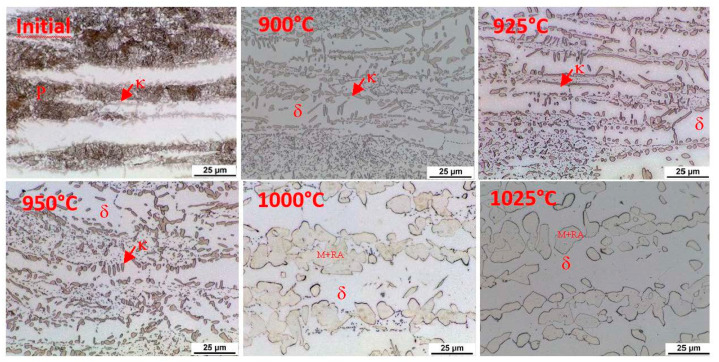
Microstructure of the initial state after different austenitisation temperatures, heat C with 7 wt% Al. P—pearlite, δ—ferrite, γ—austenite, κ—kappa phase.

**Figure 14 materials-15-02539-f014:**
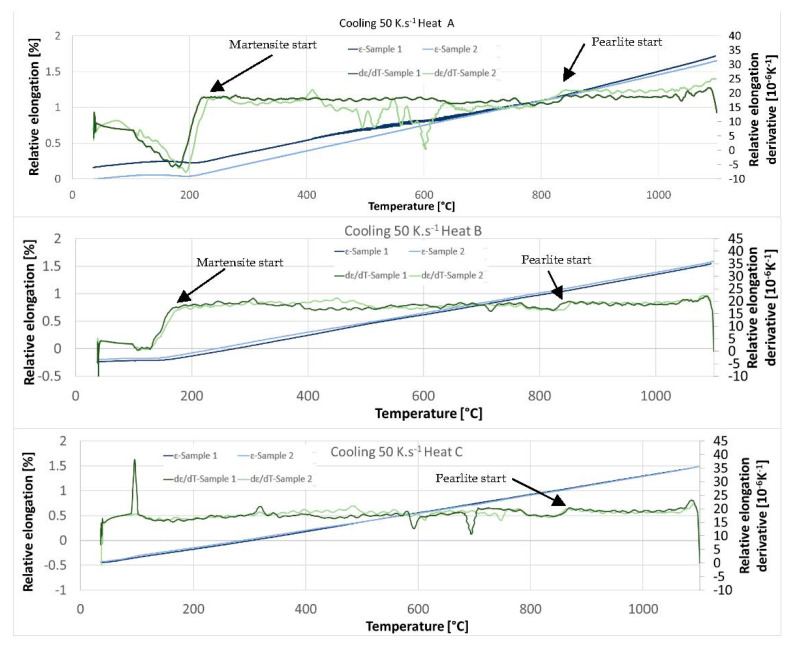
Dilatometric curves and their derivation—cooling at 50 °C s^−1^.

**Figure 15 materials-15-02539-f015:**
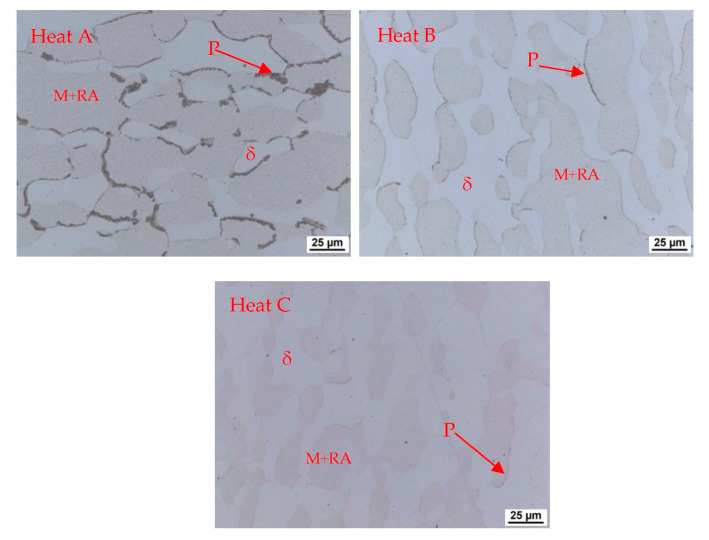
Pearlitic islands at the boundaries of the original austenitic grains formed during cooling at 50 °C s^−1^. P—pearlite, δ—ferrite, M—martensite, RA—retained austenite.

**Figure 16 materials-15-02539-f016:**
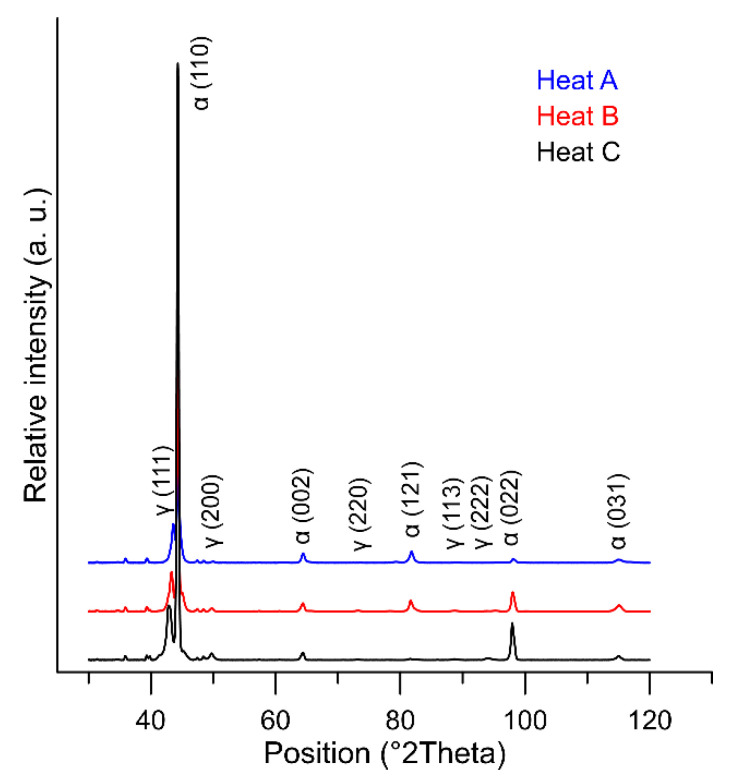
Determination of the proportion of retained austenite in the structure cooled from 1100 °C—heat C (7 wt% Al).

**Figure 17 materials-15-02539-f017:**
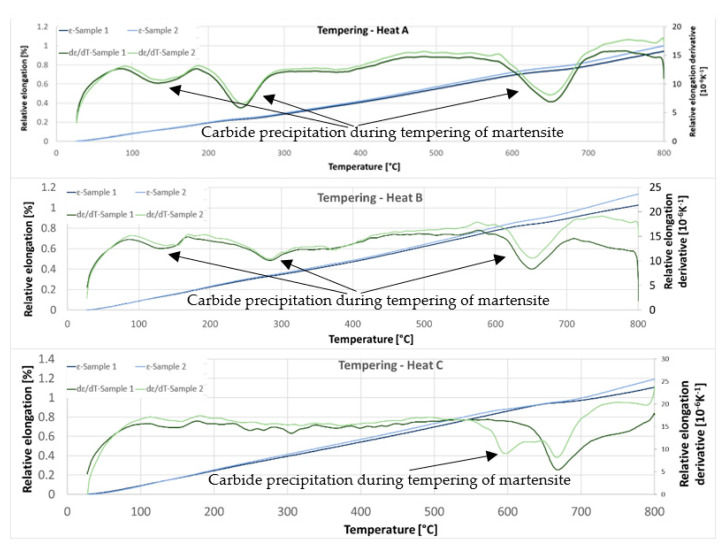
The dilatometric curves of relative elongation and derivative of relative elongation during tempering—heating 3 °C/min.

**Figure 18 materials-15-02539-f018:**
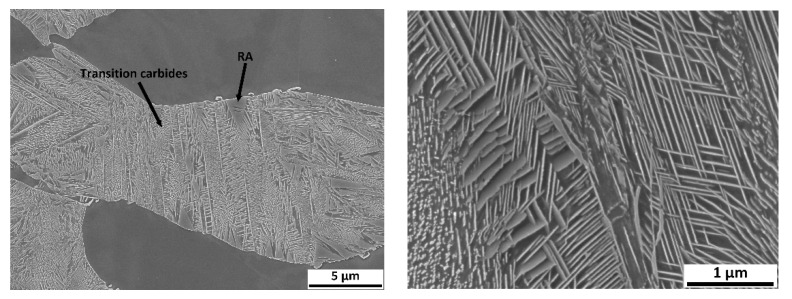
Microstructure during the tempering of heat A with 5 wt% Al. Tempering temperature 300 °C—heating rate 3 °C/min. Magnification 5000× (left) 30,000× (right).

**Figure 19 materials-15-02539-f019:**
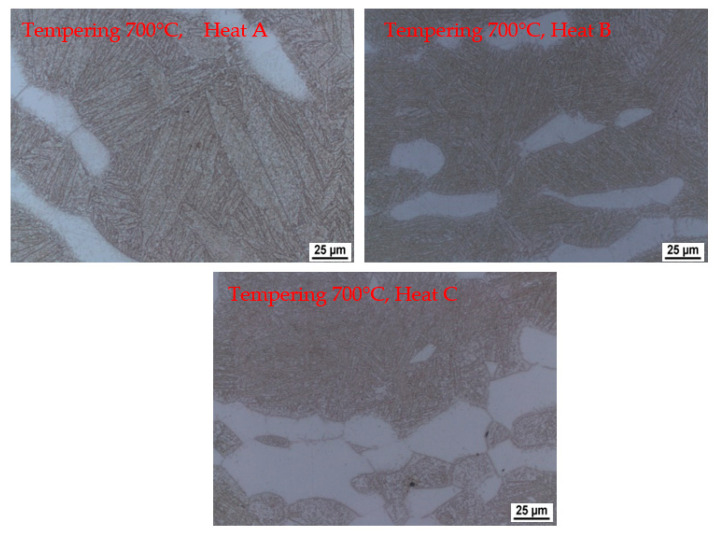
Microstructures of individual heats during tempering. Tempering temperature of 700 °C—heating rate 3 °C/min.

**Figure 20 materials-15-02539-f020:**
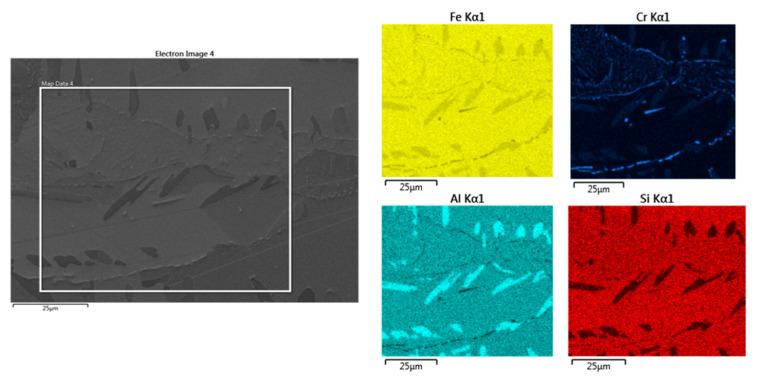
EDS analysis of the heat with 7 wt% Al.

**Figure 21 materials-15-02539-f021:**
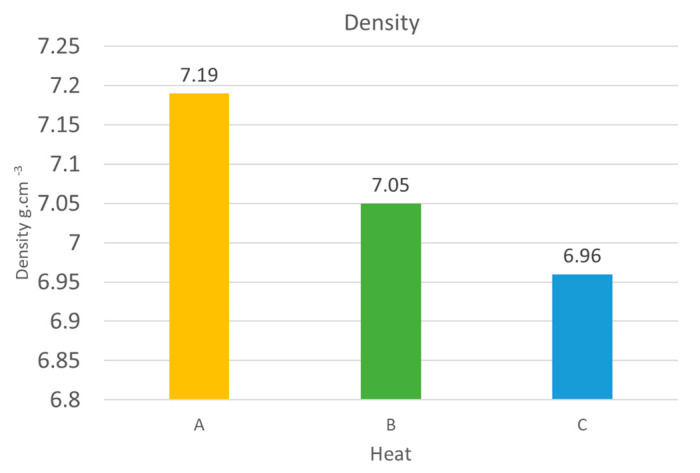
The density of individual heats.

**Table 1 materials-15-02539-t001:** Chemical composition of the heats in wt%.

Element	A	B	C
C	0.72	0.71	0.73
Si	1.45	1.45	1.43
Mn	0.11	0.11	0.11
Cr	1.99	2	2.04
Ni	0.52	0.54	0.55
Al	5.1	6.1	7.2
B	0.005	0.005	0.005

**Table 2 materials-15-02539-t002:** Amount of ferrite in the microstructure—initial state.

Ferrite Percentage		
Heat	Ar. Mean	St. Deviation
A (5 wt% Al)	33.7	3.15
B (6 wt% Al)	40.5	1.29
C (7wt% Al)	48.5	1.4

**Table 3 materials-15-02539-t003:** Amount of untransformed austenite in the structure of each heat after quenching from 1100 °C.

Heat	Retained Austenite [%]			Ar. Mean	St. Deviation
A	10.6	9.5	10.5	10.2	0.61
B	12	11.7	10.2	11.3	0.96
C	39.7	40.1	38.9	39.6	0.61

**Table 4 materials-15-02539-t004:** Results of the spot analysis of the chemical composition of individual phases or mixtures. All in wt%.

Phase	Heat/Element								
	Heat A			Heat B			Heat C		
	Al	Si	Cr	Al	Si	Cr	Al	Si	Cr
Ferrite	6.1 ± 0.05	1.7 ± 0.05	2.2 ± 0.15	7.3 ± 0.39	1.5 ± 0.21	2.5 ± 0.26	8.2 ± 0.01	1.7 ± 0.07	1.9 ± 0.24
Kappa	7.5 ± 0.53	1.4 ± 0.24	2.6 ± 0.16	9.1 ± 0.45	1.5 ± 0.26	2.6 ± 0.44	12.9 ± 0.81	0.5 ± 0.21	3.5 ± 0.49
Eutectoid	5.4 ± 0.23	1.4 ± 0.1	2.5 ± 0.26	6.8 ± 0.26	1.3 ± 0.05	3.0 ± 0.19	6.9 ± 0.23	1.4 ± 0.11	9.2 ± 2.5

## Data Availability

The raw data are not publicly available due to ongoing research.

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
