# Peer review of "A New Alloying Concept for Low-Density Steels"

_materials, 2022, doi:10.3390/ma15072539_

Round 1
Reviewer 1 Report
This paper introduces a new alloying concept for this type of steel. Low-density steels are a promising material, particularly in applications requiring low weight and good mechanical qualities. The production of low-density steels is complicated, thus the research in this topic is critical. The present paper is interesting and have a novelty in the work, however, to be accepted for publication the following comments need to be addressed
- Minor English changes are required in the revised manuscript
Abstract
- In the abstract, the aim of this paper is clear. The most significant results were presented. Overall, the abstract is well written.
Introduction
- The introduction section needs to be improved by citing new and related articles of the current journal.
Material and experimental details
- The authors are required to add more details about the used JMatPro software (company, country). In addition, the author mentioned that “JMatPro is a simulation software for calculating different properties and is focused mainly 81 on multi-component systems for industrial practice” this information is public, and we can get it from the google or the site of the program, so I recommend deleting it.
- Regarding, “The main aim was to develop steels with reduced weight; the main focus was on the aluminium content of the steel in combination with different carbon contents. The density can also be reduced by using alloying elements such as silicon and manganese.” I think these sentences should be moved to introduction. It is not suitable to state the aim of the work here, besides it is a repeated sentence.
- Please add the unit of the chemical compositions in table 1. Is it %wt, or mass percent?
- The specification of the used furnace should be added, model, company, and country. In the same manner, the specification of the used Bruker should be added.
- L135, please add the preparing procedures of the samples, chemical etching used, …... I get it in line 184, please rearrange the experimental part in right order. How to put microstructure before the preparing procedures.
- I recommend moving the microstructure investigation results (Figure 3 to Figure 10) to the results part.
Results
- The author should add the unit of the amount of ferrite in the microstructure in table 2. In addition to Table 4.
- Add the equation number in L202. Is it general equation or you own one? Please add the reference here.
- Subheading should be added to L208, 283, and 339.
- Please modify Figure 11, add legend for each line colour.
- XRD data (Figure 16) needs to be modified please add the peaks phases.
Author Response
We want to thank you for reviewing the article. All changes are marked in yellow.

Reviewer 2 Report
- The present paper proposed a new alloying concept for design low-density steels. The essence of the concept is tailoring the chemical composition of the alloys. Actually, this concept or method is quite tradition. What’s the novelty of this investigation?
- The language and structure of the present paper are quite confused. It’s hard to understand the research results by reading the present manuscript.
- Both the figures and tables in the present manuscript were not well structured. The whole paper reads like a lab report, instead of a scientific paper.
- The conclusions of the present paper were not clear.
In summary, the quality of the present manuscript cannot reach the requirements of Materials journal.
Author Response

(The authors gave the same response as above.)

Reviewer 3 Report
The paper contains serious errors in the methodology of dilatometric measurements. The paper uses a classic push rod dilatometer. The paper does not provide the dimensions of the samples accepted for dilatometric tests. As a rule, for this type of devices, they are cylinders with a diameter of 6 mm to 8 mm and a length of about 30 mm to 60 mm. A cooling speed of 50 K / s is absurd. It is not possible to ensure a cooling of such a large sample that the entire volume of the sample is cooled down at this rate. The fact that such a cooling speed can be set on the device, although it seems problematic (maybe it is 50K / min), does not mean that the test results can be interpreted correctly. As a rule, the results of dilatometric tests are limited to heating and not to interpret cooling. In exceptional cases, when we are sure that the cooling rate corresponds to the uniform cooling of the entire sample, such results can be interpreted. Very small samples of the mg order and other devices, e.g. DSC (differential scanning calorimeter) are used to study cooling processes.
Author Response

(The authors gave the same response as above.)

Reviewer 4 Report
The paper approaches a very interesting theme, one that is present interest and with certain promise, the expertise developed in the field of alloying for low-density steels.
Strong points
The title is adequate and describes very well the content of the entire paper.
The paper presents a practical approach, that is correct from a statistical and technical standpoint and uses modern instruments of investigation, that are highly precise and accurate.
The results are interesting and deeply discussed.
Weaknesses
- The abstract is very general, and there is a lot of irrelevant information. The abstract should be explained and showed the important aspects of paper. So, this abstract in the present form is unacceptable.
- The previous study is medium level, and isn't sufficient. It is recommended updated this section with new references, and compared them with your paper.
- What standards have been used for standard specimens and treatments?
- Figures of the specimens should be included in the paper.
- “Moreover, we compared the yield strength, tensile strength and density of the variants.” I did not see the results of these tests in the paper.
- As the results and discussion in this paper, it is better to compare the research result with other similar research result.
- The methods of analysis discussed should be discussed in more depth. Of course, the application of specimen stress is debatable. The authors need to clarify the heat treatments applied and the temperature control.
- The conclusions must be refined so as to highlight as well as possible the interesting experimental results.
I recommend publishing the paper, but only after a review with a minor revision of the paper’s content.
Author Response

(The authors gave the same response as above.)

Round 2
Reviewer 1 Report
The authors have addressed all my comments. the revised article can be accepted in the current form.
Author Response
Thank you very much for your comments.
Reviewer 2 Report
The present paper investigates the effect of alloying elements (Al and Cr) on the phase transformation behaviors of low density steels. The phase transformation was investigated by dilatometry. The microstructure evolution during processing (rolling, quenching and tempering) was characterized by SEM and OM. The research methods and contents of the present paper is quite general.
The biggest problem is the conclusion. The authors claimed that “The very strong effect of ferrite-forming elements is evident in the experiments, especially aluminium and chromium. We come to the significant conclusion that even at very high austenitization temperatures exceeding 1100 °C, it is impossible to leave the two-phase structure of ferrite and austenite. The amount of ferrite is 30 to 50 % of the total volume. Therefore, the alloying concept stabilizes the high amount of ferrite in the structure.” The effect of Al and Cr has been intensively studied and it is widely accepted that both Al and Cr are ferrite former and stabilizer. So what’s the significance of the present investigation? Moreover, the another conclusion is that “It can be concluded that the increased aluminium content shifts the martensite start temperature to lower temperatures. It also results in a higher amount of retained austenite in the structure”. This conclusion contradicts to the classical view that Al promotes the formation of ferrite and decrease the amounts of retained austenite. Why the present paper shows totally different results? We seriously doubt the accuracy of the present paper.
The language of the paper has been improved after revision. However, the contents and conclusion of the present paper are still far to meet the requirements of Materials journal. The novelty of the paper can not reach the level of present journal.
Author Response
Thank you very much for your comments.

Reviewer 3 Report
My comments were taken into account. The dimensions of the samples and the test equipment used should always be given. I have no more comments. I can only suggest using the abbreviation CLTE (coefficient of linear thermal expansion) in place of the derivative of the relative elongation.
Author Response
Thank you very much for your comments.